# WilsonGenAI a deep learning approach to classify pathogenic variants in Wilson Disease

**Aastha Vatsyayan**[1,2‡], **Mukesh Kumar**[1,2‡], **Bhaskar Jyoti Saikia**[1,2], **Vinod Scaria**[1,2¤]*, **Binukumar B. K.**[1,2]*

1 CSIR Institute of Genomics and Integrative Biology (CSIR-IGIB), Delhi, India, 2 Academy of Scientific and Innovative Research (AcSIR), Ghaziabad, India

¤ Current address: Vishwanath Cancer Care Foundation, Mumbai, India
‡ AV and MK contributed equally and would like to be known as joint first authors.
* vinods@igib.in (VS); binukumar@igib.in (BBK)

**Data Availability Statement:** All relevant data are within the manuscript and its Supporting information files.

**Funding:** This work was supported by the Council of Scientific and Industrial Research (CSIR)

## Abstract

### Background

Advances in Next Generation Sequencing have made rapid variant discovery and detection widely accessible. To facilitate a better understanding of the nature of these variants, American College of Medical Genetics and Genomics and the Association of Molecular Pathologists (ACMG-AMP) have issued a set of guidelines for variant classification. However, given the vast number of variants associated with any disorder, it is impossible to manually apply these guidelines to all known variants. Machine learning methodologies offer a rapid way to classify large numbers of variants, as well as variants of uncertain significance as either pathogenic or benign. Here we classify *ATP7B* genetic variants by employing ML and AI algorithms trained on our well-annotated WilsonGen dataset.

### Methods

We have trained and validated two algorithms: TabNet and XGBoost on a high-confidence dataset of manually annotated, ACMG & AMP classified variants of the *ATP7B* gene associated with Wilson's Disease.

### Results

Using an independent validation dataset of ACMG & AMP classified variants, as well as a patient set of functionally validated variants, we showed how both algorithms perform and can be used to classify large numbers of variants in clinical as well as research settings.

### Conclusion

We have created a ready to deploy tool, that can classify variants linked with Wilson's disease as pathogenic or benign, which can be utilized by both clinicians and researchers to better understand the disease through the nature of genetic variants associated with it.

[IndiGenApp Grant and OLP2301]. The funders had no role in study design, data collection and analysis, decision to publish, or preparation of the manuscript.

**Competing interests:** The authors have declared that no competing interests exist.

**Abbreviations:** ACMG-AMP, American College of Medical Genetics and Genomics and the Association of Molecular Pathologists; MCC, Matthews Correlation Coefficient; NPV, Negative Predictive Value; PPV, Positive Predictive Value; SDM, Site-Directed Mutagenesis.

## Introduction

Wilson's Disease (WD) is a rare autosomal recessive disorder characterized by the presence of pathogenic mutations in the copper-transporting *ATP7B* gene. Located on chromosome 13q14.2, *ATP7B* spans 21 exons, encoding a 1465-amino-acid copper-transporting ATPase [1]. Altered gene function in WD results in copper accumulation in the liver and brain, leading to impaired functions and movement disorders. WD patients exhibit pathogenic mutations causing reduced serum holo-ceruloplasmin production. Excessive copper deposition induces oxidative stress, contributing to clinical problems like cirrhosis and fulminant hepatitis. Neurological complications arise from copper deposits in specific brain regions, leading to movement disorders and associated symptoms [2]. This complex interplay of genetic factors and copper metabolism underscores the multisystemic impact of WD.

WD is an underdiagnosed and treatable genetic condition with an estimated worldwide prevalence of around 13.9 per 100,000, derived from known pathogenic variants [3]. Several recent publications have highlighted an estimated carrier frequency of 1 in 90 individuals [4–6]. The known prevalence and carrier frequency of WD however, are confined to a few specific populations [7, 8] while in large populations like India, they remain unexplored. This opens up a unique opportunity to understand the genetic architecture of the disease in populations rich in genetic diversity such as India.

The recent availability of a framework for the interpretation of pathogenicity of genetic variants put forward by the American College of Medical Genetics and Genomics and the Association of Molecular Pathologists (ACMG & AMP) has opened up a unique opportunity to create a standardised system for interpretation of genetic variants for clinical diagnosis and genetic counselling. To assist in a better understanding of variant pathogenicity, our group has recently put together one of the most comprehensive collections of genetic variants classified according to the ACMG & AMP Guidelines [9], in form of the WilsonGen database, a robust compilation of all publicly reported *ATP7B* variants exhaustively collected from literature and across 9 large databases [10], making it the largest, most comprehensive database of it's kind, to the best of our knowledge.

While classification according to the ACMG & AMP guidelines is time-consuming and at times limited by literature and experimental evidence to confirm the pathogenicity, a number of variants remain unclassified as variants of uncertain significance (VUS). This significantly impacts the ability to classify variants, especially from unique population groups and rare variants identified from patient cohorts.

The advent of machine learning approaches in clinical medicine have accelerated the ability to analyse and interpret medical data and have been extensively used in a number of scenarios, including the rapid classification of large numbers of variants. The widespread application of such approaches in genomics however, has been limited by the lack of gold-standard datasets for training. The availability of WilsonGen database thus provides a unique opportunity in this aspect.

Here, we describe a machine learning approach trained on a gold-standard ACMG-classified variant dataset for pathogenicity in the *ATP7B* gene for accurate classification of variants. We also use the approach for reclassification of VUS variants in public datasets so as to enable quick variant interpretation in clinical and research settings. To the best of our knowledge, ours is the only approach based on a manually ACMG-classified dataset, dedicated specifically to WD variants. A public implementation of the algorithm is available at: https://github.com/aastha-v/WilsonGenAI.

## Materials and methods

### Datasets

The variants and their pathogenicity ascertained according to the ACMG and AMP guidelines and available in the WilsonGen database were taken up for analysis. This dataset contained a total of 1458 genetic variants manually classified according to the ACMG & AMP guidelines. Non-exonic variants were removed due to lack of sufficient training data, as were VUS variants. This resulted in a variant dataset of 723 unique variants, out of which 410 were annotated as pathogenic, 167 as likely pathogenic, 9 as benign and 137 as likely benign. Fig 1 offers an overview of our entire workflow.

### Variant parameters

The variant VCF was run through the ANNOVAR [11] tool, which annotated the variants with allele frequencies (AF) from three global population and subpopulation datasets: GnomAD [12], 1000Genomes [13] and GME [14]. We further added the position of the first

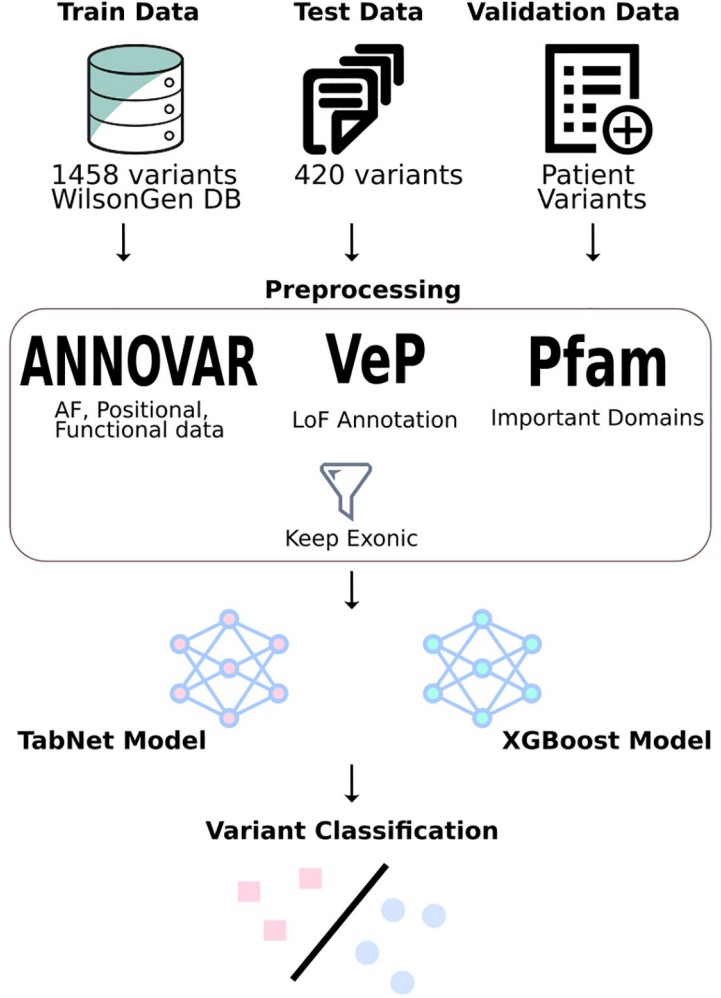

**Fig 1. The overview of the workflow followed for model development and variant classification with TabNet and XGBoost models.**

amino acid change for each variant as "Start_pro", which would offer positional data along with "Start" which marked the position of nucleotide change. Further, we added the categorical attributes "Pfam_imp_domain" that would indicate whether the variant overlapped with an important protein domain based on the Pfam database, as well as "LoF_HC_Canonical" categorical attribute which would indicate if the variant was a high-confidence loss-of-function variant present in a canonical transcript. The exonic function (e.g. frameshift insertion/deletion. Stopgain/Startloss etc.) was further encoded into numbers. Feature selection was performed manually with all features with missing values that exceeded 80% for each class being removed. Finally, all /likely pathogenic variants were encoded as "1" and all benign/likely benign variants as "0". A total of 73 attributes were thus obtained and are detailed in S1 Table.

## AI models

For our analysis, we considered two state-of-the-art deep and machine learning models, namely TabNet and XGBoost, to train on the ACMG-classified gold standard dataset.

We had previously utilized the Weka suite [15] to test the performance of several algorithms including NaiveBayes, SMO, J48, and RandomForest on the dataset available to us then, comprising of 725 variants split into 70% train—30% test datasets. Traditionally, tree ensemble models are recommended for classification and regression problems with tabular data [16]. Our results proved to be in concordance, since RandomForest and J48 outperformed others. We thus chose to work with the XGBoost classifier, which is one of the most widely used gradient-boosted decision trees, especially for tabular datasets. XGBoost is reported to perform faster and better than other models such as RandomForest for missing data, and with class imbalanced datasets. It also has in-built regularization which prevents overfitting, which models like RandomForest can be prone to. The XGBoost algorithm creates decision trees in sequential form, wherein increased weights of incorrectly predicted variables are fed into the next tree. The algorithm has been created to handle sparse data effectively, which mirrors real-world situations where data is often found to be missing or containing frequently repeating values.

Additionally, we chose to also utilize the novel deep learning neural network TabNet [17], which was specially created for tabulated datasets. TabNet has been reported to outperform tree methods including XGBoost for certain tabular datasets [18]. Unlike other deep learning models, TabNet mimics the learning of decision trees, through the use of its transformer architecture, enabling the model to quickly decipher complex data patterns. TabNet uses sequential attention to choose features at each decision step. Feature selection is done instance-wise, i.e. it could be different for each variant in the training dataset. To the best of our knowledge, this is the first implementation of TabNet for the classification of variants based on their pathogenicity.

Since the performance of the two models with respect to each other seems to vary based on datasets used [16], we decided to include results from both models for assessment.

## Hyperparameter selection and cross validation

Our models were run with different input parameters until convergence. The best performing model by accuracy was taken up. The PyTorch [19] implementation of Google's TabNet was used for model creation, while Anaconda [20] was used to enable the use of Scikit-learn, Pandas, Matplotlib and Seaborn to enable analysis and visualisation for both models.

For TabNet, SimpleImputer was used to replace missing data with a constant value. Further, the model's mask_type parameter was set to 'entmax', which showed a better overall performance than the default 'sparsemax'. The 'weights' parameter was set to '1' to address class imbalance, while the batch size was set at the maximum recommended 10% of the total data

size at 72. A maximum of 1000 epochs were allowed with a patience (i.e. the number of epochs to wait for improvement before terminating the training run) of 100.

For XGBoost, the hyper-parameters were selected and evaluated using a 5-fold cross validation (CV) approach. A randomized search on the hyperparameters was performed using RandomizedSearchCV (CV = 5). Class imbalance was corrected using the scale_pos_weight parameter set at 3.95. The following hyper-parameters were finally used (Table 1):

The mean cross_val_score function (CV = 10) was used to test model performance for both models across multiple test/train splits. Several models with and without the hyperparameters determined during tuning, were tested for performance using accuracy metrics described below. The best performing model was then selected.

## Independent validation dataset

An additional number of 420 variants were curated from published literature not included in the WilsonGen database till 2022. The variants were classified according to the ACMG & AMP guidelines as described previously. The dataset comprised of 31 variants which were annotated as pathogenic, 29 which were likely pathogenic, 96 were likely benign, and the remaining variants were classified as VUS. Thus we had a total of 156 variants in our independent test dataset.

## Accuracy estimates

The following accuracy estimates were used to evaluate the performance of the models: a) Sensitivity b) Specificity c) Accuracy d) Positive Predictive Value (PPV) e) Negative Predictive Value (NPV), and f) Matthews Correlation Coefficient (MCC). All data used in this study is freely accessible at: https://clingen.igib.res.in/WilsonGen/ The source code of both our models is available at https://github.com/aastha-v/WilsonGenAI. The models have been standardized on Ubuntu 18 LTS. The instructions and code for the preprocessing pipeline, variant classification through our models, as well as for generating one's own models are also freely included.

## Patient data validation

**Generating variants and functional validation.** *ATP7B* plasmid (pLB1080; Addgene) was subjected to site-directed mutagenesis (SDM) according to the manufacturer's instruction (Agilent, 200522) using the set of primers shown in Table 2.

To understand the impact of WT (wild type) *ATP7B* and its protein mutants, knock-out HepG2 cells were cultured under the standard conditions. Different plasmids were transfected using lipofectamine-3000 (Thermo Scientific, L3000008). Post 24 hours, cells were treated

**Table 1. Model hyperparameters used for the XGBoost model.**

| Hyperparameter | Value | Hyperparameter | Value | Hyperparameter | Value |
|---|---|---|---|---|---|
| base_score | 0.5 | gpu_id | -1 | min_child_weight | 1 |
| booster | gbtree | grow_policy | depthwise' | missing | nan |
| callbacks | None | importance_type | None | monotone_constraints | () |
| colsample_bylevel | 1 | interaction_constraints | ' | n_estimators | 50 |
| colsample_bynode | 1 | learning_rate | 0.25 | n_jobs | 0 |
| colsample_bytree | 0.9 | max_bin | 256 | num_parallel_tree | 1 |
| early_stopping_rounds | None | max_cat_to_onehot | 4 | predictor | auto |
| enable_categorical | FALSE | max_delta_step | 0 | random_state | 0 |
| eval_metric | None | max_depth | 6 | reg_alpha | 0 |
| gamma | 0.2 | max_leaves | 0 | reg_lambda | 1 |

**Table 2. Primers used in site-directed mutagenesis.**

| Variants | Forward Primer (5'—3') | Reverse Primer (5'—3') |
|---|---|---|
| c.2564C>T (S855F) | CTCCTGTGATGAGGAACTCATCAGCCATGGTATT | AATACCATGGCTGATGAGTTCCTCATCACAGGAG |
| c.813C>A (C271X) | GCCTCCGCAGTCTCCACCACAGCCA | TGGCTGTGGTGGAGACTGCGGAGGC |

with 500 μM CuCl2 for 6 hours and replaced with fresh media. After 18 hours, spent media was collected to estimate the exported copper using the manufacturer's protocol (Sigma, MAK127). The colorimetric data of the assay was analyzed using an unpaired-t-test, with a p-value<0.05 considered statistically significant for all three sets of experiments.

## Results

Both models performed best with a 70–30% train-test split. The TabNet model additionally split the 30% test set into 50% train and 50% validation subsets during training.

### Accuracy estimates

**TabNet.** Although the model was set to run at a maximum of 1000 epochs, it stopped the training at 187th epoch with the best accuracy of 99% on the validation and 97.24% on the test sets respectively. The overall MCC was 0.92. The Precision, Recall and F1 scores are shown in S2 Table. S1 Fig shows the accuracy and Fig 2 the Area Under the Curve (AUC) plot; the receiver operating characteristic curve (ROC) was 0.996. Further, S2 Fig shows the confusion matrix for our test data; out of 109 variants taken as part of the 50% test subset data, the model accurately predicted 84 as pathogenic and 22 as benign. The Precision-Recall curve is shown in S3 Fig, with the overall area under the precision-recall curve (AUPRC) determined to be 1. The model learning rate and loss are plotted in S4 Fig. Additionally, the model Specificity, and its Negative Predictive Value (NPV) were both 1.

**XGBoost.** The XGBoost model had an overall accuracy on the test set of 0.986175, AUC 0.9926 and MCC of 0.952773. Fig 2 shows the AUC plot, while S2 Fig depicts the confusion matrix. The Precision, Recall and F1 scores are shown in S2 Table. The Precision-Recall curve is shown in S3 Fig, with the overall AUPRC determined to be 1. Additionally, the model Specificity was 0.989, and its NPV was 1.

### Validation in an independent set of variants classified according to the ACMG & AMP guidelines

After removing all non-exonic variants, we had a total of 96 benign/likely benign variants clubbed together as benign, and 60 pathogenic/likely pathogenic variants clubbed together as "pathogenic". Upon running our models on the data, the TabNet model accurately classified all correctly, while XGBoost correctly classified 60 variants as pathogenic and 95 as benign, as shown in the confusion matrices in S5 Fig. Scatterplots of class probability vs the actual ACMG class for each model across all 156 variants are shown in S6 and S7 Figs for TabNet and XGBoost respectively.

### Comparison with CADD

Both our models performed better than CADD, which only had scores for 53 out of the 156 variants included in the independent ACMG-qualified test set. TabNet had an overall accuracy on the test set of 1, and XGBoost of 0.9935, while CADD only had an overall accuracy of

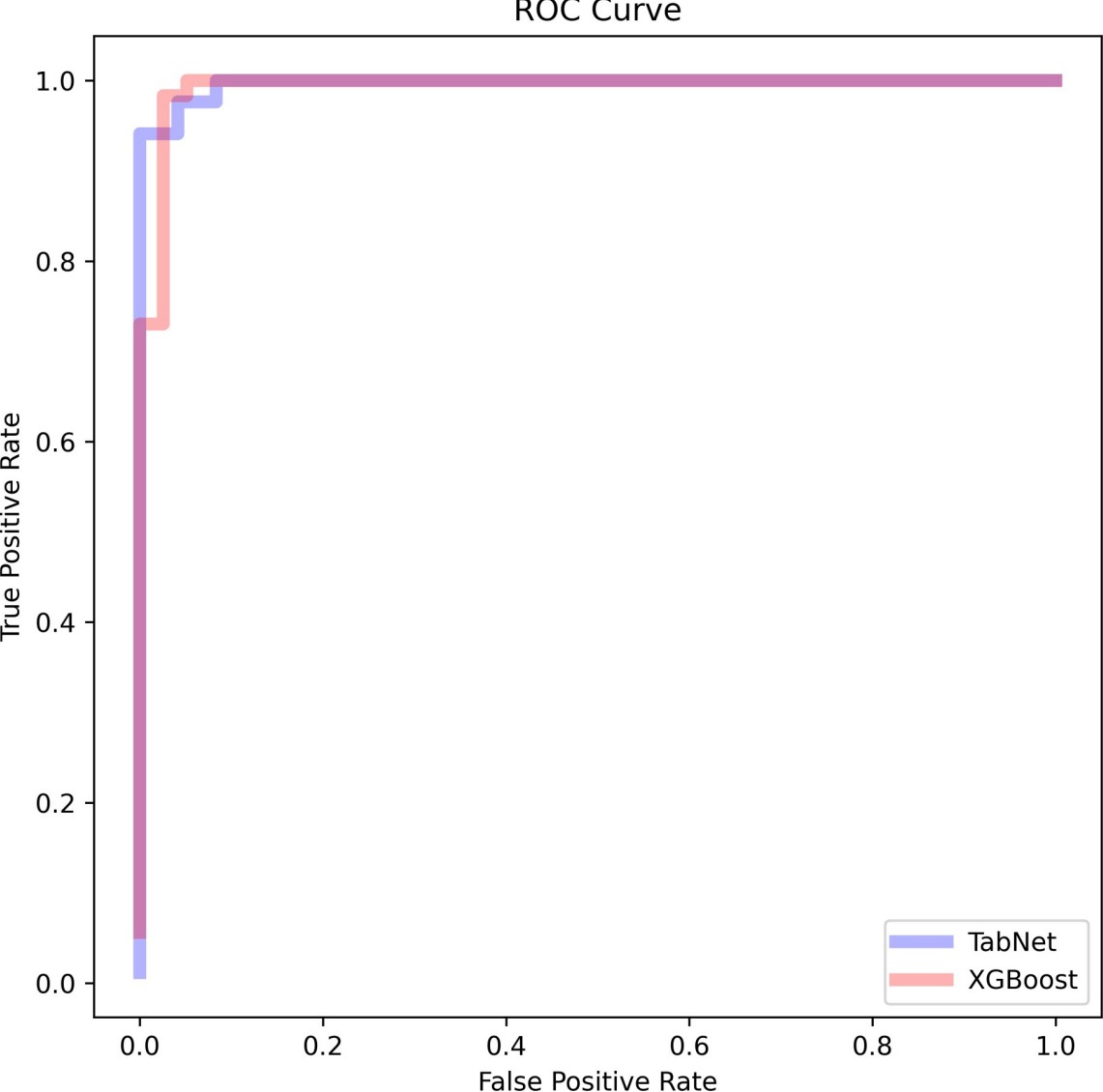

**Fig 2.** The receiver operating characteristic curve for (A) the TabNet and (B) the XGBoost model.

0.9811 on the limited number of variants it predicted. A complete comparison between the Accuracy, PPV, NPV and MCC is shown in S3 Table.

## Comparison with other models

To the best of our knowledge, ours are the only models trained on an ACMG/AMP gold standard dataset specifically created for *ATP7B* variants linked with Wilson's Disease. While other deep learning models based on ACMG/AMP guidelines such as RENOVO [21] and MLVar [22] exist, they are either not trained on manually classified variants/attributes, or do not follow a disease-specific approach. As each disease follows different genetic mechanisms, generalization for all is difficult to achieve by a single model. We have, however included scores generated by running RENOVO, as well as pre-determined scores obtained for 11 other

models including AlphaMissense, REVEL, SIFT, Polyphen2, Eigen-PC, LRT, MutationTaster, FATHMM, PROVEAN, MetaLR, and MutationAssessor [23–33] in S3 Table, and as S8 Fig. Our models were able to outperform the others over the combined metrics of Accuracy, PPV, NPV, and MCC.

## Reclassification of VUS variants

We collected all *ATP7B* variants of unknown significance and conflicting or missing classification from the ClinVar [34] database as well as our in-house data and used the model to reclassify them. Out of 977 exonic variants, TabNet reclassified 736 variants as pathogenic and 241 as benign. XGBoost on the other hand reclassified 800 as pathogenic and 177 as benign. Overall, a 91.4% concordance in predictions (726 pathogenic and 167 benign variants) was observed between the two models. The complete list of these variants and their reclassification can be accessed in S4 Table.

Scatterplots of class probability vs the predicted class for each model across 251 exonic VUS variants that were a part of our validation dataset are shown in S9 and S10 Figs for TabNet and XGBoost respectively.

## Patient data validation

**Impacts of WT *ATP7B* protein variants in cellular copper excretion.** The copper assay data for the *ATP7B* variants, S855F and C271X (positive control for impaired *ATP7B*) showed reduced copper levels in the media in comparison to the WT *ATP7B* (Fig 3). This implies that WT *ATP7B* promotes the excretion of excess copper in the media while mutants, S855F and C271X show impaired protein function.

We ran both our models on each of the variants: both models accurately identified the control C271X variant as pathogenic, and also classified S855F as pathogenic. Thus, both our models tested on functionally proven data provide accurate classifications of the variant.

## Feature importance and computational efficiency

The feature importance of the top 20 features are depicted in S11 Fig. The larger the score, the higher the impact of the feature on the model. Both models had 10 features in common, relying on Loss of function (LoF) information, wherein a genetic lesion prevents the formation of a normal gene product thereby leading to disease. They also take into account the genomic position of the mutation (Start: nucleotide), which could dictate a pathogenic effect. Additionally they rely on global prevalence of variants (1000Genomes AF -ALL)—the number of high frequency disease causing variants is usually small, i.e. most pathogenic variants are rarely prevalent. The remaining features common to both models consist of pathogenicity scores obtained from 7 prediction tools (MetaSVM, MCAP, MutPred, SIF4G, REVEL, PolyPhen2 HDIV, and MutationTaster). Additional details of these features can be seen in S1 Table.

The XGBoost model additionally relies on the exonic function of the variant (Function), i.e. the nature of the effect the variant has (a stopgain/loss variant for example, would have a larger effect on the protein than a synonymous variant). It also takes into account the allele frequencies reported in the GnomAD database, which is a larger population dataset. Finally, it also considers conservation scores (Siphy 29way logOdds and MutationAssessor) that dictate how conserved a given site is among mammals, indicating a potentially important location, and thus a potentially more disruptive effect, as well as additional pathogenicity scores (DANN, MetaRNN, and BayesDel).

The TabNet model additionally considers variant prevalence in Gnomad (GnomadAF—Raw) and the Northeast African subset of the Greater Middle East populations (GME AF—

# Media Copper Estimation

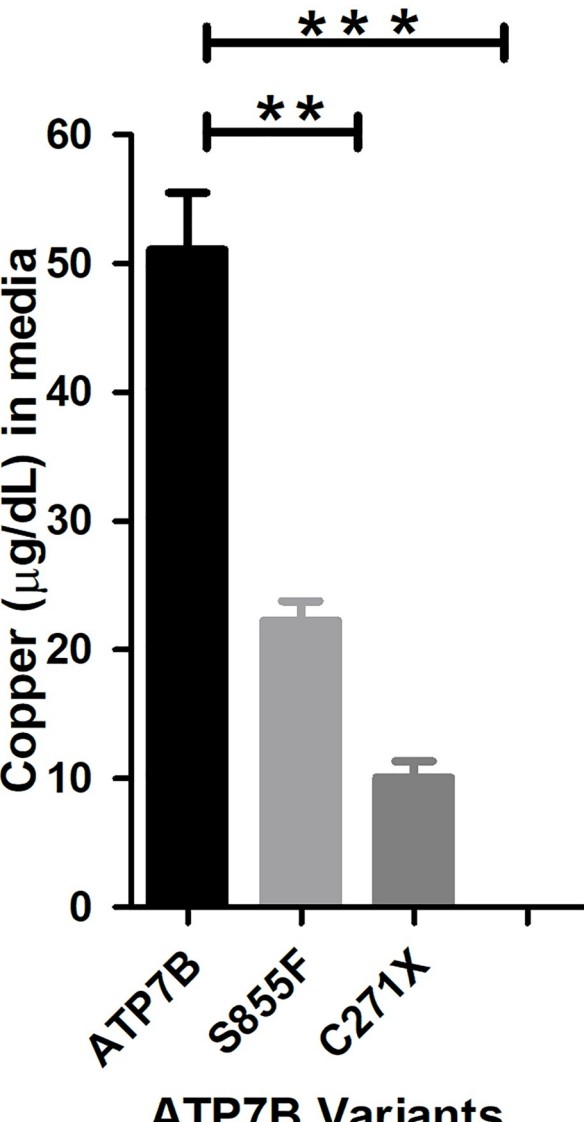

**Fig 3. Copper exposure in *ATP7B* Knock-out HepG2 cells overexpressed with the plasmid containing wildtype and mutant *ATP7B* gene.** The copper transport activity of *ATP7B* mutants S855F and C271X is significantly impaired in comparison to the wild-type *ATP7B*, where N = 3, ** denotes p value less than 0.01 and *** denotes p value less than 0.001.

NEA), as well as pathogenicity and conservation scores (LRT, integreated_fitCons, PrimateAI, Eigen-PC-raw coding, and LIST-S2).

Thus both models take a well-rounded approach, and consider different aspects that determine variant pathogenicity, and are thus able to make reliable predictions. Further, the train

dataset labels have been determined through ACMG classification that take into account all aspects of relevant biological data including functional and segregational evidence. As such the models capture patterns among the attributes that lead to these classifications.

The complete time taken to process a VCF file into suitable input, and then train a model was plotted for each model separately, and are shown in S12 Fig.

## Discussion and conclusion

In our work we have created two tools that can be used to classify variants of the *ATP7B* gene linked with Wilson's Disease. While tree-based XGBoost is one of the most reliable algorithms for tabular data, our study shows that TabNet, a deep learning model designed specifically for tabular data, slightly outperforms it in the classification of *ATP7B* variants. We have trained these models on a dataset classified through the application of ACMG guidelines, the gold standard in variant classification. Additionally, the data is a robust compilation of all publicly available variants of the gene exhaustively collected from literature and across 9 large databases. Thus the models were trained on accurately classified variants that capture all currently known types of exonic variants associated with Wilson's Disease, due to which we anticipate the models to be able to generalize to newly reported variants in the future. We have shown our models' accuracy through functional validation as well as comparison with other models. Finally, to address the large numbers of already reported variants of uncertain significance, we have collected and classified 977 exonic variants through both models; the predictions achieved a 91.4% concordance across 726 pathogenic and 167 benign variants. We have made these predictions openly available, along with their class probabilities to facilitate a better understanding of variant pathogenicity for clinicians and researchers.

Clinical diagnosis of Wilson's Disease is often challenging due to the heterogenous nature of symptoms it presents with. Genetic testing has thus been included in the diagnosis process as part of the Leipzig scoring system [23]. Additionally, testing can also rule out other genetic disorders such as some congenital disorders of glycosylation that mimic Wilson disease, but are not caused by *ATP7B* variants [35]. Since early diagnosis may prevent patients ever becoming symptomatic, infant and newborn screening, as well as family screening also become important. Accurate clinical interpretation of variants is therefore essential for diagnosis. Our models offer a means of applying learning of patterns based on classification by ACMG rapidly to a large number of variants, which otherwise is a time consuming and expertise-dependent process. Given the complex nature and varied mechanisms of genetic diseases, adopting a generalized approach to classifying causative variants is ill advised. We have shown this through the superior performance of our models over other general ACMG based models. To the best of our knowledge, no other models based on the ACMG classification of Wilson's disease variants currently exist.

We believe therefore, that our models can be utilized for the rapid classification of Wilson's Disease variants for better understanding of their pathogenicity in both research and clinical settings.

Limitations: Even though the WilsonGen database is an exhaustive compendium of currently known and classified variants, the number of classified exonic variants still remains small. ACMG classification of variants is also a time-consuming process, and thus a newer dataset may take time to make. We have thus been able to test model generalization on a dataset of 156 test variants. Additionally, the functional classification of ATP7B variants is still ongoing. Upon its completion, a clearer picture of which of the two models has performed better will be able to be obtained.

## Supporting information

**S1 Fig. Train and validation accuracies of the TabNet model across 187 epochs.**
(TIF)

**S2 Fig. Confusion matrix depicting the models' predictions on the 30% test data.** Fig A represents TabNet while B represents XGBoost.
(TIF)

**S3 Fig. The Precision-Recall curve of both the models.**
(TIF)

**S4 Fig. The model learning rate and loss of the TabNet model across187 epochs.**
(TIF)

**S5 Fig. Confusion matrix of predictions made on the ACMG-qualified independent validation dataset comprising of 156 variants.** Fig A represents TabNet while B represents XGBoost.
(TIF)

**S6 Fig. Scatterplot of class probability vs the actual ACMG class for the TabNet model across the validation set of 156 variants.**
(TIF)

**S7 Fig. Scatterplot of class probability vs the actual ACMG class for the XGBoost model across the validation set of 156 variants.**
(TIF)

**S8 Fig. Barplot comparing the accuracy, MCC, NPV and PPV of 13 models with TabNet and XGBoost.** Abbreviations: MAssessor—MutationAssessor; MTaster—MutationTaster.
(TIF)

**S9 Fig. Scatterplot of class probability vs the predicted class for the TabNet model across all VUS variants 251 exonic VUS variants that were a part of the validation dataset.**
(TIF)

**S10 Fig. Scatterplot of class probability vs the predicted class for the XGBoost model across all VUS variants 251 exonic VUS variants that were a part of the validation dataset.**
(TIF)

**S11 Fig. Plot depicting the feature importance of the top 15 features of each model.** The x-axis for XGBoost plots F-score, while that of TabNet plots scores for each feature.
(TIF)

**S12 Fig.** Plot depicting the complete time taken to process a VCF file into suitable input, and then train a model was plotted for (A) TabNet and (B) XGBoost respectively.
(TIF)

**S1 Table. A complete list of the 73 features used in training the model, along with the ACMG attribute they provide information about, along with their description, as well as their datatype.**
(XLSX)

**S2 Table.** The classification report with the Precision, Recall and F1 scores for the TabNet model (A) and XGBoost model (B) respectively.
(XLSX)

**S3 Table. Comparison of the performance of both models against 13 other models on the independent test dataset.**
(XLSX)

**S4 Table.** A list of 977 exonic variants of uncertain significance reclassified by our models TabNet (A) and XGBoost (B). Variants highlighted in bold represent concordance between predictions from both algorithms. Table (C) describes the nucleotide and protein changes in HGVS nomenclature, and also describes each variant's exonic function.
(XLSX)

## Acknowledgments

AV acknowledges a Senior Research Fellowship from ICMR. MK acknowledges a Senior Research Fellowship from CSIR.

## Author Contributions

**Conceptualization:** Vinod Scaria, Binukumar B. K.

**Data curation:** Aastha Vatsyayan, Mukesh Kumar, Bhaskar Jyoti Saikia.

**Formal analysis:** Aastha Vatsyayan, Mukesh Kumar.

**Funding acquisition:** Vinod Scaria, Binukumar B. K.

**Investigation:** Vinod Scaria, Binukumar B. K.

**Project administration:** Vinod Scaria, Binukumar B. K.

**Software:** Aastha Vatsyayan.

**Supervision:** Vinod Scaria.

**Validation:** Mukesh Kumar, Bhaskar Jyoti Saikia.

**Writing – review & editing:** Mukesh Kumar, Vinod Scaria, Binukumar B. K.

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
