## [Decision Letter · Decision Letter 0]

18 Jan 2024

PONE-D-23-35773WilsonGenAI a deep learning approach to classify pathogenic variants in Wilson DiseasePLOS ONE

Dear Dr. BK,

Thank you for submitting your manuscript to PLOS ONE. After careful consideration, we feel that it has merit but does not fully meet PLOS ONE’s publication criteria as it currently stands. Therefore, we invite you to submit a revised version of the manuscript that addresses the points raised during the review process.

We look forward to receiving your revised manuscript.

Kind regards,

Jianhong Zhou

Staff Editor

PLOS ONE

“Funding from the Council of Scientific and Industrial Research (CSIR) through the IndiGenApp Grant and OLP2301

6. We note that Figure 1 in your submission contain copyrighted images. All PLOS content is published under the Creative Commons Attribution License (CC BY 4.0), which means that the manuscript, images, and Supporting Information files will be freely available online, and any third party is permitted to access, download, copy, distribute, and use these materials in any way, even commercially, with proper attribution. For more information, see our copyright guidelines: http://journals.plos.org/plosone/s/licenses-and-copyright.

Reviewers' comments:

Reviewer's Responses to Questions

**Comments to the Author**

1. Is the manuscript technically sound, and do the data support the conclusions?

Reviewer #1: Partly

Reviewer #2: Yes

2. Has the statistical analysis been performed appropriately and rigorously? 

Reviewer #1: Yes

Reviewer #2: Yes

3. Have the authors made all data underlying the findings in their manuscript fully available?

Reviewer #1: Yes

Reviewer #2: No

4. Is the manuscript presented in an intelligible fashion and written in standard English?

Reviewer #1: Yes

Reviewer #2: Yes

5. Review Comments to the Author

Reviewer #1: This paper presents genetic variant classification using machine learning techniques, specifically TabNet and XGBoost, to classify ATP7B gene variants associated with Wilson's Disease. The study's strength lies in its robust training and validation on a high-confidence dataset and its practical application, as evidenced by successful independent verification and potential utility in clinical and research settings. I have several comments that are need to be addressed.

Major:

(1) Why were TabNet and XGBoost chosen as the primary models for this analysis over other deep learning or machine learning models? What specific advantages do they offer for this type of data and problem? Please provide the comparison with other relevant deep learning methods.

(2) The authors mentioned that TabNet uses sequential attention for feature selection, which is instance-wise. How does this impact the generalizability of the model across different datasets or variants? Is there a risk of overfitting to specific features in the training dataset?

(3) The authors note that XGBoost is effective in handling sparse data. However, it is not clear that on how this capability was specifically advantageous in current study, given the characteristics of used dataset?

(4) For the models, the authors have set specific hyperparameters. The manuscript need more details about how were these parameters chosen.

(5) The authors adjusted the scale_pos_weight in XGBoost for class imbalance. How significant was the class imbalance in used dataset, and how did this adjustment impact the model's performance, especially in terms of precision and recall?

(6) TabNet stopped training at the 187th epoch out of a possible 1000. Was this due to an early stopping criterion based on validation accuracy? A big epoch size does not necessarily increase the accuracy of the model. How was the risk of overfitting addressed given the excessive epoch size (>100)?

(7) The authors mentioned the top 20 features in feature importance plots for both models. Could the authors provide insights into what these features represent and how they contribute to the pathogenicity classification? How interpretable are these models in terms of understanding the biological significance of these features?

(8) The manuscript needs more details on how specificity, negative predictive value (NPV), or area under the precision-recall curve (AUPRC) considered?

(9) The test sets' composition (number of benign vs. pathogenic variants) and their source (whether they were balanced or reflective of real-world distributions) are not detailed. How might this affect the models' generalizability to other datasets or real-world scenarios?

(10) The comparison with CADD and other models like RENOVO and MLVar suggests superior performance of your models. However, were these comparisons made under similar conditions (e.g., same datasets, metrics)? How do the models compare in terms of computational efficiency and scalability?

(11) When reclassifying variants of uncertain significance, how did the authors validate the accuracy of these reclassifications? Is there a risk of introducing bias or errors in this process, given the uncertain nature of these variants?

(12) The discussion section of the manuscript needs to be significantly expanded. These are few points the authors may consider while revising the discussion section. In discussion the authors should interpret and explain the findings, placing them in the context of the broader field. Begin by summarizing the main findings of the study, highlighting how they address the research questions or hypotheses stated in the introduction. Then, contextualize these results within the existing literature, discussing how these findings align with or differ from previous research and the potential reasons for these similarities or differences. What is the significance and implications of the results, considering both their theoretical and practical applications. Acknowledge the limitations, discussing how they might affect your findings and suggesting areas for future research to address these gaps. This section should bridge the gap between the presented research and the larger scientific community, demonstrating how this work contributes to and advances the field.

Minor:

(1) It would help readers to introduce Wilson's disease in the introduction section.

(2) The relevance of choosing the ATP7B gene needs to be added in the introduction.

(3) "Non-exonic variants and VUS were removed from the analysis and this resulted in a variant dataset of 723 unique variants, ..." Explain why. What is VUS? Expand all the abbreviations at the first use.

(4) lines 106 – 113: The parameters could be presented in a table.

Reviewer #2: Summary:

Vatsyayan et al. applied two ML models (TabNet and XGBoost) to classify ATP7B genetic variants of Wilsen disease based on highly engineered features of each variant. Both models show very high classification accuracy, indicating the potential usability to reduce the manual evaluation efforts such as following guidelines of American College of Medical Genetics and Genomics and the Association of Molecular Pathologists. However, because of the lack of comparison with other variants classification methods e.g. disease agnostic model, it is hard to tell the novelty of WilsonGenAI and whether WilsonGenAI really adds value to Wilson's disease specific variant classification. Due to the high requirement of storage size of the WilsonGenAI, I have not evaluate the software itself. Please see the following comments for major revision:

Major:

1. In introduction, please review and discuss related works. Line 174-180 should be part of the introduction.

2. In results, in addition to CADD, please compare the WilsonGenAI results with more state-of-the-art methods such as Eigen-PC, REVEL, AphaMissense, etc. Moreover, the argument of not comparing the proposed methods with RENOVO and MLVar are not convincing. Please also include these results in Table S3. Without seeing these baseline results, it is difficult to conclude TabNet and XGBoost are necessary Wilson's Disease specific model. The model comparison figure (e.g. barplot of Table S3) might be the main figure highlighted by this paper.

3. line 68, there are much more pathogenic variants than benign class. Have the authors considered whether the imbalanced distribution will influence the results?

4. line 74-76, it seems the three population used for annotation is different from the population of WilsonGen dataset. Can the authors discuss more on the potential problem of this inconsistency?

5. Figure S1. Can the authors show both training and validation loss in order to easily see whether the model is overfitting or not.

6. Figure S2. It seems the important features identied by XGBoost and TabNet are quite different but their ROC are similar. Can the authors discuss more about this?

7. Figure S3. It seems the accuracies are very unstable. Can the authors comment on this problem?

8. Figrue S4. It is weird that the total number of variants are different between the methods.

9. Since the proposed models only consider two classes, in practice, for the variants with around 0.5 predicted probability, should the user regard them as VUS? Is there a recommended threshold? This is especially important as the authors claimed WilsonGenAI could be used for clinical diagnosis.

10. For the VUS of independent datasets (line 124), is the predicted score of them around the margin of the two classes? Is there a trend or correlation between the predicted score and the 5 ordinal classes?

11. Line 189-200, is there a specific consideration to choose S855 and C271X for validation? Ideally, it would be very interesting to see if some VUS with very high predicted pathogenic probability can be validated to lead to low Copper concentration.

12. It seems the independent dataset is not available.

Minor:

1. Please define abbreviation WD, VUS before using it.

2. line 149. Please round the number.

6. PLOS authors have the option to publish the peer review history of their article (what does this mean?). If published, this will include your full peer review and any attached files.

Reviewer #1: No

Reviewer #2: No

---

## [Author Response · Author response to Decision Letter 0]

5 Mar 2024

A point-by-point response to reviewers

Reviewer #1: This paper presents genetic variant classification using machine learning techniques, specifically TabNet and XGBoost, to classify ATP7B gene variants associated with Wilson's Disease. The study's strength lies in its robust training and validation on a high-confidence dataset and its practical application, as evidenced by successful independent verification and potential utility in clinical and research settings. I have several comments that are need to be addressed.

Major:

(Q1) Why were TabNet and XGBoost chosen as the primary models for this analysis over other deep learning or machine learning models? What specific advantages do they offer for this type of data and problem? Please provide the comparison with other relevant deep learning methods.

Response: We appreciate the inquiry from the reviewer. In our initial exploration of model selection, we conducted a comprehensive analysis using the Weka suite (Witten et al. 2011). The dataset under consideration at that time comprised 725 variants, split into 70% training and 30% testing datasets. The table below illustrates the train and test accuracies achieved by different algorithms:

Model Train Accuracy Test Accuracy

RandomForest 97.925 98.611

J48 97.41 98.61

SMO 96.89 97.92

NaiveBayes 83.59 76.39

Consistent with conventional wisdom, tree ensemble models demonstrated superior performance on tabular data. Notably, RandomForest and J48 outperformed other algorithms. Therefore, we opted for the XGBoost classifier, a widely-used gradient-boosted decision tree, known for its efficiency in handling tabular datasets. XGBoost offers advantages such as faster execution, robust performance with missing data, and effective handling of class-imbalanced datasets. Its built-in regularization helps mitigate overfitting, a concern associated with models like RandomForest.

Furthermore, recognizing the specialized nature of tabular data, we incorporated the TabNet deep learning model into our analysis. TabNet, designed specifically for tabulated datasets, leverages a transformer architecture to emulate the learning of decision trees. This design enables TabNet to rapidly discern intricate data patterns. While XGBoost excels in certain scenarios, TabNet has demonstrated superiority over tree methods for specific tabular datasets.

Given the variability in the comparative performance of these models depending on the dataset, we decided to include results from both XGBoost and TabNet for a comprehensive evaluation. This dual-model approach allows for a more nuanced understanding and robust assessment of the predictions made by each model.

(Q2) The authors mentioned that TabNet uses sequential attention for feature selection, which is instance-wise. How does this impact the generalizability of the model across different datasets or variants? Is there a risk of overfitting to specific features in the training dataset?

Response: We appreciate the reviewer's insightful consideration of the generalizability of TabNet across diverse datasets. While TabNet incorporates features like prior scales to mitigate overfitting, predicting the exact extent of model generalization remains challenging, particularly in the absence of ample accurately classified variant datasets. Our training utilized the most extensive dataset of Wilson's Disease variants reported in literature, encompassing nine large datasets with ACMG classifications. The model consistently demonstrated high classification accuracy across both classes, as assessed by Matthews Correlation Coefficient (MCC). This performance instills confidence in its potential to perform well on other real-world datasets. Regrettably, we were unable to conduct additional testing due to the scarcity of available ACMG-classified or functionally validated variant datasets. Despite this limitation, our rigorous training on a diverse and comprehensive dataset enhances our confidence in the model's ability to generalize to different datasets and variants. Future investigations and validations with additional variant datasets would certainly contribute to a more comprehensive understanding of TabNet's generalizability across a broader spectrum of genetic variations.

(Q3) The authors note that XGBoost is effective in handling sparse data. However, it is not clear that on how this capability was specifically advantageous in current study, given the characteristics of used dataset?

Response: We appreciate the reviewer's observation, and would like to clarify the specific advantage of XGBoost's capability to handle sparse data in our study. Real-world datasets often exhibit missing values, posing a challenge for deep learning models. In our dataset, certain features, such as pathogenicity and conservation scores, had missing values due to the inherent characteristics of their respective prediction algorithms. To address this issue with TabNet, we had to perform imputation by substituting missing data with a constant value far outside the scale of all scores. This substitution aimed to avoid introducing unintended bias.

Contrastingly, XGBoost demonstrated an inherent advantage in handling sparse data. Unlike TabNet, XGBoost required no data substitution for missing values, resulting in a more streamlined preprocessing step. Moreover, XGBoost's Sparsity-aware Split Finding algorithm automatically determines optimal splits for data points with missing values, contributing to an improved overall performance. This capability proved advantageous in our study, in terms of streamlining the preprocessing phase and enhancing the model's efficiency in handling sparse data patterns.

(Q4) For the models, the authors have set specific hyperparameters. The manuscript need more details about how were these parameters chosen.

Response: We appreciate the reviewer's request for more details on the hyperparameter selection process. In the TabNet model, we explored different values for the mask_type parameter during experimentation. The "entmax" setting demonstrated superior overall prediction accuracy compared to the default "sparsemax," leading us to choose it for model training. The “patience” parameter, governing the number of epochs to await improvement before terminating a training run, was set at 100, with a maximum of 1000 epochs allowed. Various dataset splits were tested, including 70% train and 30% test, as well as 80% train and 20% test, to ensure robust testing.

For the XGBoost model, hyperparameters were carefully selected and evaluated using a 5-fold cross-validation approach. A randomized search on hyperparameters was conducted using RandomizedSearchCV with 5-fold cross-validation. To address class imbalance, the scale_pos_weight parameter was determined by dividing the number of majority class entries by the number of minority class entries. Model performance was assessed using the mean cross_val_score function with a 10-fold cross-validation. Multiple models, with and without the determined hyperparameters (including scale_pos_weight), were tested using accuracy, AUC, and MCC metrics. Additionally, various train/test splits were explored to identify the best-performing model.

These details have been incorporated into the revised manuscript to provide a comprehensive understanding of the hyperparameter selection process for both the TabNet and XGBoost models.

(Q5) The authors adjusted the scale_pos_weight in XGBoost for class imbalance. How significant was the class imbalance in used dataset, and how did this adjustment impact the model's performance, especially in terms of precision and recall?

Response: Our train set had 577 pathogenic/likely pathogenic and 146 benign/likely benign variants. Given this imbalance, we adjusted the scale_pos_weight parameter to the recommended value of 3.95. To evaluate the impact of this adjustment, we employed the Matthews Correlation Coefficient (MCC) metric, which comprehensively considers all components of the confusion matrix, namely true positives (TP), true negatives (TN), false positives (FP), and false negatives (FN). It thus enabled us to determine if the classifier was doing well on both positive and negative classes. This was pertinent due to the potential clinical implications associated with misclassifying benign variants as pathogenic.

Comparing models with and without the adjusted scale_pos_weight, we observed an improvement in performance with the weighted model achieving an MCC of 0.95 compared to 0.90 without the weights. While precision remained consistent at 0.98 for both models, the weighted model exhibited an enhanced recall of 1.00 as opposed to 0.98 without the adjustment. Moreover, the F1-score demonstrated improvement with the weighted model, reaching 0.99 compared to 0.98 without the adjustment.

It is noteworthy that our pursuit of optimal hyperparameter configurations involved experimenting with various combinations. Throughout this process, models consistently performed better when the scale_pos_weight parameter was appropriately adjusted. This underscores the significance of addressing class imbalance, as reflected in the superior performance and robustness of models that incorporated the weighted approach.

(Q6) TabNet stopped training at the 187th epoch out of a possible 1000. Was this due to an early stopping criterion based on validation accuracy? A big epoch size does not necessarily increase the accuracy of the model. How was the risk of overfitting addressed given the excessive epoch size (>100)?

Response: Indeed, TabNet implemented an early stopping mechanism based on the validation accuracy metric during training. The early stopping criterion was defined by setting patience at 100, meaning that if the accuracy did not improve for 100 consecutive epochs, the training process would halt. Subsequently, TabNet automatically selected the epoch with the best accuracy score for making predictions on the evaluation set.

 To mitigate the risk of overfitting, the model width, representing the number of nodes in a layer, was set to 8. Additionally, the parameter n_steps was configured to 3. These decisions aimed to strike a balance between model complexity and generalization capacity, reducing the likelihood of overfitting.

Furthermore, to validate the robustness of the model, its performance was rigorously assessed on an independent validation set, where it was able to correctly classify all variants across both classes. We thus anticipate the model to be able to generalize well on data beyond the training set. 

(Q7) The authors mentioned the top 20 features in feature importance plots for both models. Could the authors provide insights into what these features represent and how they contribute to the pathogenicity classification? How interpretable are these models in terms of understanding the biological significance of these features?

Response: Certainly, the feature importance plots for both models shed light on the factors influencing pathogenicity classification, derived from a comprehensive training set of 73 attributes. These attributes include variant positional information, global population prevalence, pathogenicity prediction scores from various tools, and evolutionary conservation scores.

The top features identified by both models highlight critical determinants of pathogenicity. Loss of function (LoF) information emerges as a key contributor, emphasizing the significance of genetic lesions that impede normal gene product formation, a hallmark of disease causation. The genomic position of the mutation(Start: nucleotide), could also be important in predicting pathogenic effect. Further, the global prevalence of variants, as indicated by the 1000Genomes allele frequency (1000Genomes AF - ALL), underscores the observation that the number of high frequency disease causing variants is usually small, i.e. most pathogenic variants are rarely prevalent across a population. The remaining features common to both models are pathogenicity scores from seven prediction tools (MetaSVM, MCAP, MutPred, SIF4G, REVEL, PolyPhen2 HDIV, and MutationTaster), reflecting the amalgamation of diverse computational predictions.

The XGBoost model additionally considers exonic function (Function), which describes the nature of the effect the variant has (a stopgain/loss variant for example, would have a larger effect on the protein than a synonymous variant).Allele frequencies from the GnomAD database, representing a larger population dataset, are also considered.It also takes into account conservation scores (Siphy 29way logOdds and MutationAssessor) that dictate how conserved a given site is among mammals, indicating a potentially important location, and thus a potentially more disruptive effect. The model further incorporates pathogenicity scores from DANN, MetaRNN, and BayesDel.

The TabNet model additionally considersvariant prevalence across Gnomad (GnomadAF - Raw) and the Northeast African subset of the Greater Middle East populations (GME AF - NEA). Pathogenicity and conservation scores, including LRT, integrated_fitCons, PrimateAI, Eigen-PC-raw coding, and LIST-S2, enhance the model's ability to capture nuances in variant significance.

Thus both models take a well-rounded approach, and consider different aspects that determine variant pathogenicity, and are thus able to make reliable predictions. Further, the train dataset labels have been determined through ACMG classification that take into account all aspects of relevant biological data including functional and segregational evidence. As such the models capture patterns among the attributes that lead to these classifications.

The table below is a subset of Supplementary Table 1, and offers greater detail on each of the top 20 important features:

Feature Name ACMG2015 Description Dtype

Function PVS1, BP7 Exonic function of the variant (e.g.: nonsynonymous SNV, stopgain/loss, frameshift insertion/deletion etc.) category

DANN Scores PP3, BP4 Deleterious Annotation of genetic variants using Neural Networks. Score range: 0-1. float64

Start: Nucleotide Genomic location of nucleotide int64

MetaSVM PP3, BP4 A radial SVM model to predict pathogenicity, trained on whole exome variants. Score range: -2 to 3. float64

Siphy 29way logOdds Scores PP3, BP4 SiPhy score based on 29 mammals genomes. The larger the score, the more conserved the site. Score range: 0 to 37.9718. float64

Gnomad AF - ALL BA1, BS1, BS2, PM2 Alt allele Frequency in the GnomAD database float64

LoF PVS1, BP7 Whether a variant is High Confidence LoF category

MCAP Scores PP3, BP4 Pathogenicity classifier for rare missense variants in the human genome. Score range: 0-1. float64

MutPred Scores PP3, BP4 Automates the inference of molecular mechanisms of disease from amino acid substitutions. Models changes of structural features and functional sites between wild-type and mutant protein sequences float64

SIFT4G Score PP3, BP4 Faster implementation of SIFT for wider range of organisms. Score range: 0-1. float64

1000Genomes AF -ALL BA1, BS1, BS2, PM2 Allele frequency in the 1000 Genomes database float64

MetaRNN Scores PP3, BP4 Pathogenicity prediction scores for human nonsynonymous SNVs (nsSNVs) and non-frameshift (NF) indels. float64

BayesDel with AF Scores PP3, BP4 Deleteriousness meta-score for coding and non-coding variants, SNVs and small insertion / deletions from database with integrated MaxAF. Score range: -1.29334 to 0.75731. float64

REVEL Scores PP3, BP4 Predicting the pathogenicity of missense variants on the basis of individual tools: MutPred, FATHMM, VEST, PolyPhen, SIFT, PROVEAN, MutationAssessor, MutationTaster, LRT, GERP, SiPhy, phyloP, and phastCons. Score range: 0-1. float64

MutationAssessor Scores PP3, BP4 Predicts the functional impact of amino-acid substitutions in proteins, such as mutations discovered in cancer or missense polymorphisms. The functional impact is assessed based on evolutionary conservation of the affected amino acid in protein homologs. Score range: -5.545 to 5.975. float64

Polyphen2 HDIV Scores PP3, BP4 Polyphen2 prediction based on HumDiv; The PolyPhen-2 score predicts the possible impact of an amino acid substitution on the structure and function of a human protein. Score range: 0-1.

---

## [Editor Report · Decision Letter 1]

1 May 2024

WilsonGenAI a deep learning approach to classify pathogenic variants in Wilson Disease

PONE-D-23-35773R1

Dear Dr. BK,

We’re pleased to inform you that your manuscript has been judged scientifically suitable for publication and will be formally accepted for publication once it meets all outstanding technical requirements.

Kind regards,

Muhammad Salman Bashir, M.S.C

Academic Editor

PLOS ONE
---

## [Editor Report · Acceptance letter]

7 May 2024

PONE-D-23-35773R1 

PLOS ONE

Dear Dr. BK, 

I'm pleased to inform you that your manuscript has been deemed suitable for publication in PLOS ONE. Congratulations! Your manuscript is now being handed over to our production team.

Kind regards, 

on behalf of

Dr. Muhammad Salman Bashir 

Academic Editor

PLOS ONE